# Fault interpretation in seismic reflection data: an experiment analysing the impact of conceptual model anchoring and vertical exaggeration

Juan Alcalde[1,2], Clare E. Bond[2], Gareth Johnson[3], Armelle Kloppenburg[4], Oriol Ferrer[5], Rebecca Bell[6], Puy Ayarza[7]

[1] Department of Structure and Dynamics of the Earth, Institute of Earth Sciences Jaume Almera, ICTJA-CSIC, Lluis Sole i Sabaris s/n, 08028 Barcelona, Spain
[2] Department of Geology and Petroleum Geology, School of Geosciences, University of Aberdeen, Aberdeen AB24 3UE, UK
[3] Department of Civil and Environmental Engineering, University of Strathclyde, Glasgow, G1 1XZ, UK.
[4] 4DGeo, Daal en Bergselaan 80, 2565 AH The Hague, The Netherlands
[5] Institut de Recerca Geomodels, Departament de Dinàmica de la Terra i de l'Oceà, Facultat de Ciències de la Terra, Universitat de Barcelona, Barcelona, c/ Martí i Franquès s/n., 08028, Spain
[6] Basins Research Group (BRG), Department of Earth Science & Engineering, Imperial College, Prince Consort Road, London, SW7 2BP, UK
[7] Department of Geology, University of Salamanca, 37008 Salamanca, Spain

*Correspondence to*: Juan Alcalde (juan.alcalde@abdn.ac.uk)

**Abstract.**

The use of conceptual models is essential in the interpretation of reflection seismic data. It allows interpreters to make geological sense of seismic data which carries inherent uncertainty. However, conceptual models can create powerful anchors that prevent interpreters from reassessing and adapting their interpretations as part of the interpretation process, which can subsequently lead to flawed or erroneous outcomes. It is therefore critical to understand how conceptual models are generated and applied to reduce unwanted effects in interpretation results. Here we have tested how interpretation of vertically exaggerated seismic data influenced the creation and adoption of the conceptual models of 161 participants in a paper-based interpretation experiment. Participants were asked to interpret a series of faults and a horizon, off-set by those faults, in a seismic section. The seismic section was randomly presented to the participants with different horizontal-vertical exaggeration (1:4 or 1:2). Statistical analysis of the results indicates that early anchoring to specific conceptual models had the most impact on interpretation outcome; with the degree of vertical exaggeration having a subdued influence. Three different conceptual models were adopted by participants, constrained by initial observations of the seismic data. Interpreted fault dip angles show no evidence of other constraint (e.g. from the application of accepted fault dip models). Our results provide evidence of biases in interpretation of uncertain geological and geophysical data, including the use of heuristics to form initial conceptual models and anchoring to these models, confirming the need for increased understanding and mitigation of these biases to improve interpretation outcomes.

## 1 Introduction

Reflection seismic data is used to image and understand the subsurface structure of the earth, across scales and tectonic settings (e.g. Park et al., 2002; Simancas et al., 2003; Martí et al., 2008). As with other geophysical methods, seismic images are indirect representations of complex changes in the physical properties of rocks in the subsurface. Seismic images therefore carry inherent uncertainty related to the geological architecture, but also to the acquisition, processing and visualisation methods applied to the seismic data. These uncertainties combined make seismic images subject to multiple geological interpretations or, in other words, non-unique solutions (Frodeman, 1995; Rankey and Mitchell, 2003; Bond et al., 2007; Saltus and Blakely, 2011).

Geoscientists employ mental models (or "conceptual models") that integrate their observations and that conform to their understanding of the world (Shipley and Tikoff, 2016). When confronted with geological data, interpreters need to apply different conceptual models, acquired during their training and past experience (through learning), together with robust interpretation methodologies, in order to produce interpretations that honour the data, particularly in areas of great uncertainty (Bond et al., 2007; Bond et al., 2015). Interpreters need to be able to identify the key elements (e.g. growth geometries, regional level) and employ different validation techniques (e.g. balancing or restoration) that allow differentiating between (a priori similar) conceptual models (Bond, 2015). The conceptual models therefore incorporate all the elements that shape the knowledge of the geologist of a certain aspect of the geology; for example, the conceptual model of a turbidite system will include characteristics about their origin and evolution, common stratigraphic sequences, lithological composition, stratigraphic structures associated, etc. These conceptual models are dynamically modified or renewed with the arrival of new observations (input information), and are used to produce predictions (inferences) that can help to answer questions about the world (Shipley and Tikoff, 2017). Conceptual models are therefore the basis of the interpretation, as they provide the necessary criteria to make sense of the data (Frodeman, 1995).

To deal with uncertainty, interpreters employ heuristics (or 'rules of thumb') in the process of generating the conceptual models, and that makes them subject to a broad range of cognitive biases (Kahneman et al., 1982). One of these biases is related to the capability of interpreters to adjust their interpretations from their initial ideas or conceptual models. This type of bias, called anchoring, has been identified in many decision-making processes since it was first described by Tversky and Kahneman (1974), and takes place in the seismic interpretation process. Rankey and Mitchell (2003) investigated the effect of anchoring in an interpretation experiment by asking interpreters to reassess their seismic interpretations after being provided with additional well data. Their work shows that most interpreters did not feel that their interpretations needed to change substantially, in spite of data showing changes in porosity and net-to-gross predictions that did not fit with their initial interpretations. Their results suggest that interpreters were anchored to their initial conceptual models, and that they were reluctant to change their mind in light of new data. In a different experiment, Bond et al. (2007) observed that participants asked for the geographical location of the section and suggested that interpreters could use this information to build their conceptual models, by using geographically specific knowledge of e.g. the relevant tectonic setting to anchor their

interpretation. For example, an interpreter knowing a seismic section was from the North Sea may assume a conceptual model based on an extensional tectonic regime and will consciously and unconsciously look for normal faults in the seismic data. However, if the conceptual model is wrong, e.g. there is significant inversion in the seismic section, the interpretation could be compromised. So although conceptual models can be dynamically modified or renewed with the arrival of new observations,

as described by Shipley and Tikoff (2017) and others, anchoring bias often results in limited adjustment from initial models. Thus, although conceptual models are needed to develop geologically sound interpretations, they can also create anchors to potentially erroneous outcomes.

The use of tectono-sedimentary conceptual models in seismic interpretation has been extensively documented in literature (e.g. Strecker et al., 1999; Nielsen et al., 2008; Alcalde et al., 2014). Understanding what elements influence conceptual model

development and application in seismic interpretations is useful to better grasp how interpretations are made. Applying the appropriate conceptual models requires assessment, by the interpreter, of objective uncertainty, such as considering errors in data processing or acquisition, and of subjective elements, such as the potential biases they bring to the interpretation from their background and experience (Bond, 2015). Alcalde et al., (2017a) argue that image presentation also has a subdued effect in the way seismic image data is perceived and interpreted. Here, we develop this theme investigating how presentation of

vertically exaggerated seismic image data influences conceptual model choice and application, and the subsequent interpretation outcome.

Modern computer-based methods provide important advantages to the interpretation of seismic data, such as the generation of 3D models, attribute analysis or the easy access to multiple display options (e.g. change in scales, colour palettes). However, the use of computers generally results in the onscreen interpretation of a vertically exaggerated seismic image, due to the

conflicting ratios of a 1:1 seismic image with screen dimensions (Bond, 2015). Furthermore, most 2D seismic cross-sections published in literature are displayed vertically exaggerated (Stewart, 2011), although it is likely that multiple displays were employed during the interpretation stage. Vertical exaggeration of seismic image data creates images with apparent reflection continuity and exaggerates dips of structures and horizons. Conscious application of seismic image stretching is used in the seismic interpretation process because it helps to enhance certain aspects of the display that ease the interpretation (Stewart,

2011). It helps for instance to amplify low relief structures, that appear otherwise compressed and difficult to differentiate (Feagin, 1981; Bertram and Milton, 1996). For example, Brothers et al. (2009) report that vertical exaggeration helped them to delineate small changes in stratal geometry, otherwise imperceptible, in their seismic interpretation study of the Salton Sea. Vertical exaggeration can also be used to mitigate the difference between vertical and horizontal sampling, which can be considerable depending on the acquisition parameters, the impact of which is to make images appear stretched (Stewart, 2011).

These examples highlight the usefulness of scale variation during interpretation.

However, changes in appearance of seismic image data through, sub-conscious or conscious, vertical exaggeration change an interpreter's perception of an image. The change in image character is often unintentional, and can result in unwanted perceptual bias during interpretation, and subsequently lead to misinterpretations, particularly if the interpreted geological structures are complex (Stone, 1991). Vertical exaggeration can also make features, like gas escape chimneys, appear narrower

than they are (Horozal et al. 2009). Black et al. (1994) noticed that vertically exaggerated seismic sections can result in gently dipping reflections being perceived as more steeply dipping; which may lead to the erroneous conclusion that migration of the seismic data is required. Similarly, Stewart (2012) investigated the impact of vertical exaggeration on fault dip and observed that structural restoration of interpretations conducted in exaggerated sections lead to unrealistic subsurface models. Thus,
vertical exaggeration in seismic interpretation can have positive and negative influences on interpreter perception of the image and interpretation outcome.

Here we test the theory that the presentation of seismic image data in a vertically exaggerated format impacts the perceptions of interpreters, influencing the conceptual models they apply in their interpretation and their outcome. We focus on analysis of fault and horizon interpretations in a clipped seismic image. Interpreters were randomly presented with different vertical
exaggerations (1:2 and 1:4) of the same seismic image. Statistical analysis of fault and horizon placement, fault dip angle, fault dip direction and fault type, allow us to draw conclusions on the effect of vertical exaggeration on interpretation.

## 2 Experiment set up

The interpretation experiment consisted of a c. 15 km long clipped portion from a 2D seismic image from the Browse Basin, NW Australia (Figure 1) available on the *Virtual Seismic Atlas* (www.seismicatlas.org). This seismic image has been
interpreted as a series of normal faults dipping to the NW (left hand-side of the section) overlain by post-tectonic sediments, These faults could potentially have been formed in the Late Carboniferous to Early Permian rifting event (Struckmeyer et al., 1998; Keep and Moss, 2000). The area has undergone different stages of reactivation since the Early Triassic, so inversion structures can also be found (Keep and Moss, 2000).

The section used in this experiment was originally downloaded with no vertical exaggeration (i.e. with an approximate
horizontal to vertical ratio of 1:1), according to the *Virtual Seismic Atlas* information. In a series of interpretation experiments, the seismic image was presented to participants with horizontal to vertical exaggeration of 1:4 (Figure 2a) or 1:2 (Figure 2b), hereafter called *1:4* and *1:2* sections. The sections were presented in two-way traveltime (TWT) and no information about the actual depth of the sections was provided. The participants were asked to "interpret the main faults crossing the section as deep as possible", as well as to add a "sedimentary horizon to mark the displacement", and were given 15-30 minutes to complete
their interpretations. The experiment as presented to the participants can be found in the Supplementary Information.

The participants also completed an anonymous questionnaire designed to collect information about their background, training, knowledge and experience in structural geology and seismic interpretation. The interpretation experiment was completed by 161 students of which 126 participants (78% of the total) were undergraduate students and 35 participants (22% of the total) were postgraduate students, from different universities in the UK, France and Spain. The participants have mostly geology
(72.5%) and geophysics (12.5%) backgrounds and considered themselves as having basic to good proficiency in structural geology and seismic interpretation (>93% of the participants). We focused this experiment on students only to observe the potential variability in interpretation of the same section in a group of people with similar experience and background.

## 3 Interpretation results

The two vertically exaggerated seismic images were assigned randomly to the participants: the *1:2* section was interpreted 88 times (55%) and the *1:4* section 72 times (45%). The interpretation results were digitised manually and then converted to a 1:1 vertical exaggeration (VE=1:1) for comparison; therefore, the fault dip angles presented in this work are VE=1:1 in time. As the sections were interpreted in TWT, the analysed dips of the faults are not true dips (i.e. these observed in sections in depth), but their relative differences are still comparable. Individual examples of the interpretation results after digitisation from both the *1:2* and *1:4* sections are shown in Figure 3.

Initially, interpretations were grouped based on fault dip direction, to the left or to the right. Those interpretations with faults dipping in both directions (15 interpretations, 9.4% of the total interpretations), e.g. systems of faults and their conjugates, blank or equivocal interpretations were not included in further analyses. Of the remaining 119 interpretations, most participants interpreted faults dipping to the right (67 interpretations, 56% of the total interpretations), rather than to the left (52 interpretations, 44% of the total) (Figure 4). The relative proportion is greater in the *1:4* sections (39 interpretationsto the right, 59%) compared to the *1:2* sections (28 interpretations to the right, 53%). These two groupings were identified as it was apparent that participants interpreting faults dipping to the right and those interpreting faults dipping to the left had employed two different conceptual models to the data. This resulted in four datasets with two pairs of properties (i.e. 1:2-left, notified as '1:2L', 1:2-right or '1:2R', 1:4-left or '1:4L', and 1:4-right '1:4R') that were further analysed in detail. This subdivision allows us to study if the potential differences can be attributed to the section interpreted (i.e. *1:2* or *1:4*), or to the conceptual model used in the interpretation.

We analysed the fault type (i.e. normal or reverse) and measured the fault dip angle interpreted by the participants. The fault type results do not show significant differences between the *1:2* and *1:4* section interpretations, with 32-33% of the participants interpreting reverse faults and 67-68% interpreting normal faults (Figure 4). However, difference in fault type can be correlated to the dip-direction of the fault (Figure 5). Only one participant (3%) amongst the left-ward dipping datasets (i.e. 1:2L and 1:4L) interpreted the fault motion as reverse, while the vast majority (35 participants, 97% of the total) interpreted leftward-dipping normal faults. In contrast, most right-ward dipping faults were interpreted as reverse (31 interpretations, 56%) instead of normal (24 interpretations, 44%). This result is more pronounced in the 1:4R, with 61% of faults interpreted as reverse (14 interpretations), compared to the 53% in the 1:2R (17 interpretations).

The dip angles of the faults were calculated by drawing a horizontal line at the approximate mid-depth point (1.1 ms TWT) of the seismic section, with the aim of crossing the majority of the faults around their midpoint. Similar numbers of fault interpretations were made on the *1:4* section (a total 300 faults interpreted by 72 participants, over 4 faults interpreted per participant), and the *1:2* section (272 faults by 88 participants, over 3 faults interpreted per participant) (Figure 6). The fault dip angle analyses were compared across the four datasets (Figure 7). The largest difference between the *1:4* and *1:2* sections is highlighted here, with the median dip angle of faults of 22° in the right-ward dipping, reverse *1:4* section vs 16° in the *1:2* section (Figures 7c and 7d). The differences in normal interpretations, either left-ward (Figure 7a and 7b) or right-ward dipping

faults (Figures 7e and 7f), show only differences of 2-3°, and therefore are less conclusive. The fault dip of the only participant interpreting left-ward dipping, reverse faults was 23° on average, slightly higher than the other two groups.

To check if other factors, specifically: educational background and experience, were influencing interpretation outcome we also analysed the data for disparities between different University cohorts and between undergraduate and postgraduate students. There are no major differences in the analysed results across student cohorts from different Universities, or between undergraduate and postgraduate students. For the latter cohort the difference in numbers (undergraduate (126) vs postgraduate (35) students) is large and does not allow easy comparison; despite this the ratios of leftward and rightward dipping faults and the sense of off-set is consistent across the cohorts. The effect of level of education and experience in seismic interpretation has been raised in the past (e.g. Bond et al., 2012; Alcalde et al., 2017b) and we suggest that this is still an area of interest for future work.

## 4 Discussion

### 1. Conceptual model anchoring

Analysis of participants' interpretations shows that fault interpretations in the seismic image fall into three main categories (Figure 3): (1) left-ward dipping normal faults with right dipping horizons (Figure 3b); (2) right-ward dipping thrust faults with right-dipping horizons (Figure 3c); and (3) right-ward dipping normal faults with left-dipping horizons (Figure 3d). Only one interpretation showed left-ward dipping faults with left-dipping horizons and marked the motion of the faults as reverse (Figure 5). In addition, this interpretation did not show any evidence of correlating horizons across the fault and simply used arrows to mark the motion instead. The low number of interpretations of this type (one) and the difficulty in correlation suggests that interpreting left-dipping faults with reverse fault motions is largely impossible, given the reflection seismic characteristics of the data.

Faults and horizons (red and blue lines in Figure 3, respectively) are interpreted in three ways: (1) along left-dipping discontinuous and chaotic reflections, these align with breaks in right-ward dipping reflections that together give the appearance of a left-ward dipping chaotic seismic fabric' (Figure 3b); (2) along 'packages' of right-dipping reflections with greater continuity (Figure 3c); and (3) at an angle to these right-dipping reflections where reflection continuity is less strong (Figure 3d). Irrespective of the vertical exaggeration of the seismic image interpreted, most participants interpreted faults dipping right-ward instead of left-ward (Figure 4). At the same time, the majority of right-ward dipping faults (56%) were interpreted as reverse, in contrast to left-ward dipping faults, which are mostly interpreted as normal (97%) (Figure 7). We suggest that this is as a consequence of the seismic reflection characteristics of the different features that are being interpreted as faults and horizons. The continuity of the right-ward dipping reflections makes them a more 'certain' interpretation than the left-dipping fabric. When the right-ward dipping reflections are interpreted as horizons, leaving the left-dipping fabric to be interpreted as faults, this invariably leads to interpretation of faults with normal offsets due to the angular relationship between the fault and horizon interpretations and potentially due to the participants interpretation, consciously or sub-

consciously, of the nature and geometries of the basin sediments above (Figure 3b). When the right-ward dipping reflections are interpreted as faults, the sedimentary packages are harder to interpret and horizon interpretations are often forced to cut reflections (Figure 3d). When participants have interpreted faults at an angle to the right-ward dipping reflections, where reflection continuity is less strong, this results in steeper fault dip angles, and interpreters often interpret the right-ward dipping reflections as sedimentary packages in horsts between reverse faults (Figures 3c and 7).

In summary, from the analysis of the fault and horizon interpretations of participants, three conceptual models are identified (Figure 3) that have been applied in interpretations of the data. What we do not know is how the individual participants honed onto their 'chosen' conceptual model. The participants were prompted to interpret the faults as their main task in the experiment instructions, and as a secondary element to interpret a horizon to show fault motion; an interpretation sequence as shown in figure 9. We should state that we cannot be sure that all participants followed this workflow, but we have no evidence to suggest that they did not. Irrespective of the exact interpretation sequence, we suggest that once participants started interpreting certain 'features' in the reflection seismic image data as faults or horizons, they became anchored to an initial conceptual model and fitted the rest of their interpretation to this model. Consequently, we suggest that interpreters were likely anchored to their initial thoughts on the direction of dip of the faults and the rest of their interpretation is determined by this initial fault model, irrespective of whether later interpretative elements conform to the data e.g. horizons cutting reflections, as seen in Figure 3, this has previously been reported by Rankey and Mitchell (2003) and Torvela and Bond (2011). Although, there appears to be a threshold of tolerance for data dis-confirmation. Note that no left-ward dipping faults with a reverse sense of motion have been interpreted, in which horizons would very distinctively have cut seismic reflectors (see figure 9d).

Experience and knowledge are expected to have played a key role in informing the initial observations that led to selection of a conceptual model at the beginning of the interpretation. We purposely chose a student only cohort to mitigate against the competing effects of experience and knowledge with other factors we wanted to test. To ensure this was the case we have analysed the data for differences in interpretation outcome between students from different Universities and between undergraduate and postgraduate students. This analysis shows no strong evidence that experience had an effect on interpretation outcome.

## 2. Fault dip variability

Although we purport that the impact of conceptual model application and anchoring to models has the greatest influence on the interpretation outcomes of this experiment, the experiment results show certain differences in fault dip direction and dip angle between the *1:2* and *1:4* vertically exaggerated section interpretations (Figures 4, 6 and 7). Figure 8 shows a projection of the interpreted fault dip angles and their median values for both the *1:2* and *1:4* sections on a graph of exaggerated vs unexaggerated dip angles. The interpreted dip angles are projected onto the corresponding curves of vertical exaggeration to show the equivalent unexaggerated dip angle. The same faults interpreted in sections with differing vertical exaggeration should have the same un-exaggerated dip angle (x-axis), but a differing exaggerated dip angle (y-axis). This is the case for the median of the right-ward and left-ward dipping normal fault interpretations (magenta and dark blue circles in Figure 8,

respectively). In contrast, the median fault dip angle of the right-ward dipping reverse interpretations in the *1:2* and *1:4* sections (dark pink circles in Figure 8) are not aligned vertically, indicating that the two cohorts, i.e. participants interpreting the *1:2* and *1:4* sections, did not interpret the same dipping features as reverse faults. Interpretations of right-ward dipping faults (at least these interpreted as reverse motion) show an apparent impact of vertical exaggeration on interpretation outcome, whereas

the left-ward dipping normal fault interpretations do not. In the *1:2* section, interpretations of left-ward dipping faults have higher dip angles on average than those interpreted in the *1:4* section, and a greater spread in fault dip angle (Figure 6e and 6f).

The observations of fault dip angle and motion consistency suggest that those interpreting normal faults (either right-ward or left-ward dipping) were unaffected by vertical exaggeration. Note that the interpreted median right-ward dipping fault dip

angles are low, 15-17°; when these separated into normal and reverse faults, the right-ward dipping normal faults are very low angle 10-13° (Figure 7e-f), with the reverse faults having higher average dip angles of 16-22° (Figure 7c-d).

We did not provide the velocity model for the section used, but just for comparison, we converted the faults from TWT to depth assuming a seismic velocity of 3000 ms-1 for the area (following the assumptions and caveats outlined in Stewart, 2011) (table 1). For the reverse motion faults, the resulting dip angles in depth (31-33°) are closer to an Andersonian-predicted reverse

fault dip of (30°) and falling within the range of common reverse fault dips of 10°-30° (Anderson, 1905; 1951). The normal fault angles (14-30°), however, do not conform to predicted Andersonian fault dips of 45-60°, that are predominant in teaching materials (Alcalde et al., 2017c). The participants did not have access to the regional seismic line, that would have provided context for such low angle normal faults, nor to the actual depth of the sections, so participants may have been expected to attempt to interpret faults with higher dip angles to conform to accepted dip models of normal faults. We see no evidence of

this and interpret this observation as data and conceptual model co-confirmation acting dominantly over other reasoning (if any took place).

**Table 1: median values in two-way traveltime and their depth-converted equivalent of the 1:2 and 1:4 sections, divided by dip direction and fault motion. The dips were depth-converted using a uniform velocity model of 3000 m/s (as per Stewart, 2011).**

| Section | 1:2 | | | 1:4 | | |
|---|---|---|---|---|---|---|
| Dip direction | Right | Right | Left | Right | Right | Left |
| Fault motion | Normal | Reverse | Normal | Normal | Reverse | Normal |
| Median (TWT) | 13° | 23° | 16° | 10° | 21° | 22° |
| Median (depth-converted) | 19° | 33° | 23° | 14° | 31° | 30° |

For the interpretations of left-ward dipping faults, the extent of the vertical exaggeration of the interpreted seismic image appears to have an impact on interpretation outcome. Analysis of fault dip angle from the left-ward dipping fault interpretations of the 1:2 seismic section show a greater range in fault dip angle (standard deviation SD=16°) and a higher median fault dip angle of 29°, compared to the 1:4 section interpretations with an median dip angle of 21°, SD=13° (Figure 6e-f), that is, an 8°

higher median fault dip in the 1:2 section. If we now consider only the participants' interpretations that had also interpreted a horizon showing fault motion (Figure 7a & b), the difference in fault dip angle between the 1:2 and 1:4 sections decreases to

only 2º, with similar standard deviations of 14º and 13º. We suggest that the differences observed between the 1:2 and 1:4 sections are dominated more by erroneous seismic interpretations than by vertical exaggeration, with those making 'dubious' left-ward dipping fault interpretations not completing horizon interpretations. Similarly, for the right-ward dipping fault interpretations normal fault dip angles are low 24º-27º, but not as low as those interpreted to the right, suggesting that the faults are defined more by their seismic character than by any effects of vertical exaggeration. Testing with more display options (e.g. 1:6 or 1:8 vertical exaggeration) could be helpful to confirm this finding, and would be interesting lines for further enquiry. If we consider the observations described in the light of our knowledge of the perceptual impact of vertically exaggerated seismic images (e.g. Stone, 1991; Black et al. 1994; Horozal et al. 2009; Stewart, 2012), the *1:4* section should perceptually have better reflection continuity due to data compression (Stewart, 2011). The higher apparent reflection continuity in the *1:4* section could make the right-ward dipping reflections appear more dominant and the discontinuities between the sediment packages less dominant and narrower. The smaller range in dip angles for the *1:4* section compared to the *1:2* section (SD=14º vs 16º, respectively, Figure 6a, b) may be the result of this perceptual change. But the lack of consistency in this observation when the data is split between right-ward and left-ward dipping faults (Figure 6) and also into normal and reverse faults (Figure 7), leads us to conclude that vertical exaggeration has a subdued impact. Our interpretation of these observations is that the seismic data and conceptual model have a more dominant influence on interpretation than any perceptual bias resulting from vertical exaggeration.

Our work does not provide evidence, in this case, to support the conclusions of Stone (1991), Black et al. (1994), Stewart (2011 and 2012) that vertically exaggerated seismic sections causes perceptual bias, compared with the dominant effect of anchoring to conceptual models. We still suggest however, that multiple visualisations of the data should be made, including at a scale of 1:1 and that care should be taken when interpretations of seismic image data have been made in a vertically exaggerated form. Other experimental work (Alcalde et al. 2017b) showed that interpreters and interpretation outcomes were influenced by seismic reflection contrast and continuity, factors that can be enhanced in vertically exaggerated seismic images. We suggest that future work should further investigate the effect of vertical exaggeration on seismic image properties and interpretation outcomes.

**5 Conclusions and recommendations**

We have shown in an interpretation exercise by 161 participants that:

1. Conceptual models have greater dominance on the interpretation outcome than perceptual bias from interpreting vertically exaggerated seismic sections.

2. Initial conceptual models are anchored to and there is no evidence for reassessment by participants when data does not conform to their initial model.

3. When conceptual models are confirmed, at least initially, by the data, there is no evidence that accepted models, for example in fault dip, have an impact on interpretation outcome, and that variability in interpretation (e.g. fault dips in

our experiment) is minimal even if it does not conform to accepted models (e.g. Andersonian dips). Instead, the data drives the interpreted fault dip, and the conceptual model and data co-confirm each other.

Our results support the conclusions of other workers (Rankey and Mitchell, 2003; Bond et al. 2007; 2008) that seismic interpreters need to be aware of potential biases when interpreting seismic image data particularly in the application of conceptual models; and of the high likelihood of anchoring to initial conceptual models even when data does not confirm or conform to the model. Research has shown that awareness of biases (e.g. George et al., 2000) can help mitigate the potential impacts of bias. Thus, seismic interpreters and their employers should employ bias awareness in their interpretation workflows, and obtain multiple opinions to test a broader range of conceptual models (see Bond et al., 2008 for workflow ideas; for reasoning tests to avoid anchoring see Bond, 2015; and Macrae et al., 2016; and for the potential impact of single conceptual models on decision making see Richards et al., 2015). Research into the effectiveness of different bias awareness techniques and their impact in geological interpretation is an obvious focus for future research.

The work presented here and that of many of the authors referenced provides evidence for biases in interpretation of geological and geophysical data. The resultant interpretation outcomes are not only based on uncertain data, but these uncertainties are compounded by interpretation biases including using heuristics to form initial conceptual models and anchoring to these. Understanding how to better mitigate bias in interpretation and the competing impacts on outcomes of different biases remains a significant challenge in the geosciences.

**Competing interests**

The authors declare that they have no conflict of interests.

**Author contributions**

Both JA and CEB conceptualised and designed the interpretation experiment. CEB, AK, OF, RB and PA conducted the experiments at the University of Aberdeen, the University of Grenoble, University of Barcelona, Imperial College of London and University of Salamanca. JA was responsible for the project administration and data analyses. CEB and JA were responsible for the writing, reviewing and editing of the manuscript with help from GJ, AK, OF, RB and PA.

**Acknowledgements**

We would like to thank both Charlotte Botter and Javier Tamara for their detailed review and suggestions to improve this paper. Ignacio Marzán is also thanked for his help with the figure designs.

**Financial support**

Alcalde completed the work presented whilst supported through a NERC industry partnership grant (NE/M007251/1) and is currently funded by EIT Raw Materials – SIT4ME project (17024). Bond is currently funded through a Royal Society of

Edinburgh research sabbatical on uncertainty in seismic image interpretation. Johnson is funded by the University of Strathclyde Faculty of Engineering. Ferrer has been supported by the SALCONBELT Project (CGL2017-85532-P), the Geomodels Research Institute and the Grup de Geodinàmica i Anàlisi de Conques (2017SGR-596). Ayarza is funded by the the Regional Government of Castilla y Leon (project SA065P17).

**Special issue statement**

This article is part of the special issue "Understanding the unknowns: the impact of uncertainty in the geosciences".

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

**Figure captions**

**Figure 1: Regional seismic image from the Browse Basin (NW Australia). The black box marks the area of the section used in the interpretation experiment. Note that the vertical exaggeration of the image is high (1:8). The full section, with approximately no vertical exaggeration (i.e. 1:1 display) can be downloaded from the VSA website (www.vsa.org).**

**Figure 2: Seismic sections used in the interpretation experiment with a) 1:4 vertical exaggeration and b) 1:2 vertical exaggeration.**

**Figure 3: The seismic section and sketch interpretations of the three interpretation models applied in interpretations of the seismic section (a) by participants. b) left-dipping normal faults (in red) with right-dipping horizons (in blue); c) right-dipping normal faults with left-dipping horizons; and d) right-dipping reverse faults with right-dipping horizons.**

**Figure 4: Statistics for the interpreted fault directions (left 'L' or right 'R'), and motions (normal 'N' or reverse 'R') in the sections**
**with 1:4 and 1:2 vertical exaggeration ('VE'). The number of participants is given in brackets. Note that ambiguous interpretations (e.g. left + right-dipping fault interpretations, or no faults interpreted), corresponding to 41 interpreters (25.6% of the total), were excluded from the count.**

**Figure 5: Statistics for the interpreted fault directions (left 'L' or right 'R'), and motions (normal 'N' or reverse 'R'), separated by vertical exaggeration ('VE') 1:2 or 1:4.**

**Figure 6: Rose diagrams showing the dips of interpreted faults. Fault dips interpreted at a vertical exaggeration of: a) 1:4, b) 1:2, c) 1:4 dipping right-ward ("R") d) 1:2 dipping right-ward, e) 1:4 dipping left-ward ("L") and e) 1:2 dipping left-ward. The 'n' marks the number of faults analysed. 'SD' stands for standard deviation.**

**Figure 7: Rose diagrams showing the dips of interpreted faults and their motion. Fault dips interpreted at a vertical exaggeration of: a) 1:4, left-ward dipping and normal, b) 1:2, left-ward dipping and normal, c) 1:4 right-ward dipping and reverse, d) 1:2 right-**
**ward dipping and reverse, e) 1:4 right-ward dipping and normal, f) 1:2 right-ward dipping and normal. Note that there are fewer faults presented here than in Figure 6 due to fewer participants interpreting the fault motion. The "n" marks the number of faults analysed. "SD" stands for standard deviation.**

**Figure 8: Graph adapted from Stewart (2011) showing exaggerated and un-exaggerated dip values for all fault interpretations, showing the average fault dips for left-ward and right-ward dipping faults interpreted at 1:2 and 1:4 vertical exaggeration. The**
**medians of the dip direction and fault motion sub cohorts are also presented in the 1:2 and 1:4 curves.**

**Figure 9: Proposed interpretation sequence. (a) The seismic images were presented in both 1:2 and 1:4 vertical exaggerations. (b) Independently of the image interpreted, the participants of the experiment faced the problem of how to interpret the right-ward dipping structures and the chaotic seismic fabric. (c) Participants interpreted the central fabric either as a left-ward (blue) or right-ward (orange) dipping fault, which consequently triggered (d) the horizon interpretation determining the motion (normal, green**
**horizons; and reverse, pink horizons) of the fault. The left-ward dipping, reverse faulting interpretation (crossed out in red) is too difficult to fit to the seismic data, and so only one participant chose this interpretation.**

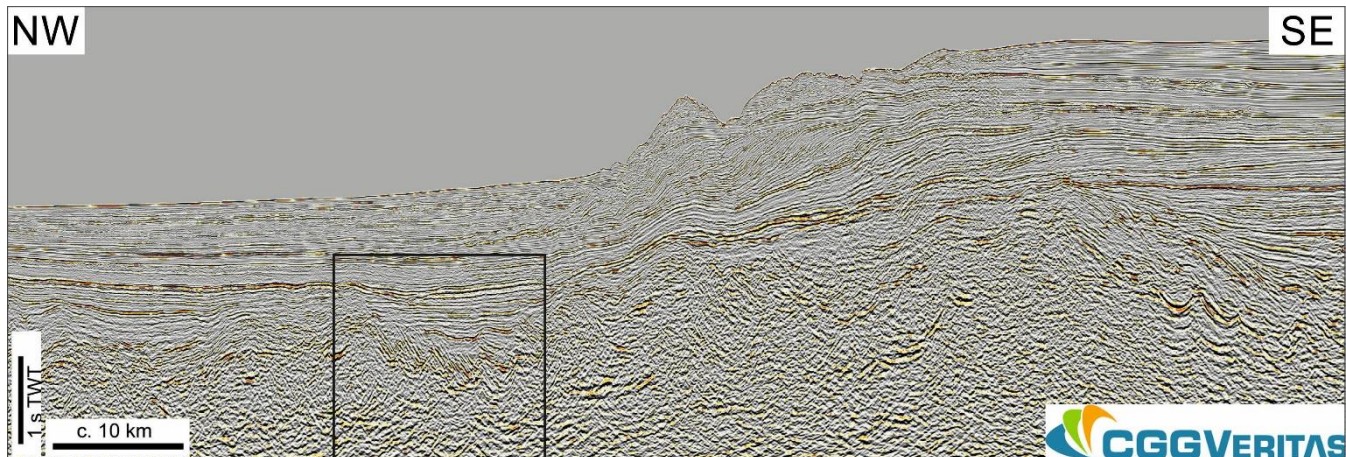

**Figure 1**

a)

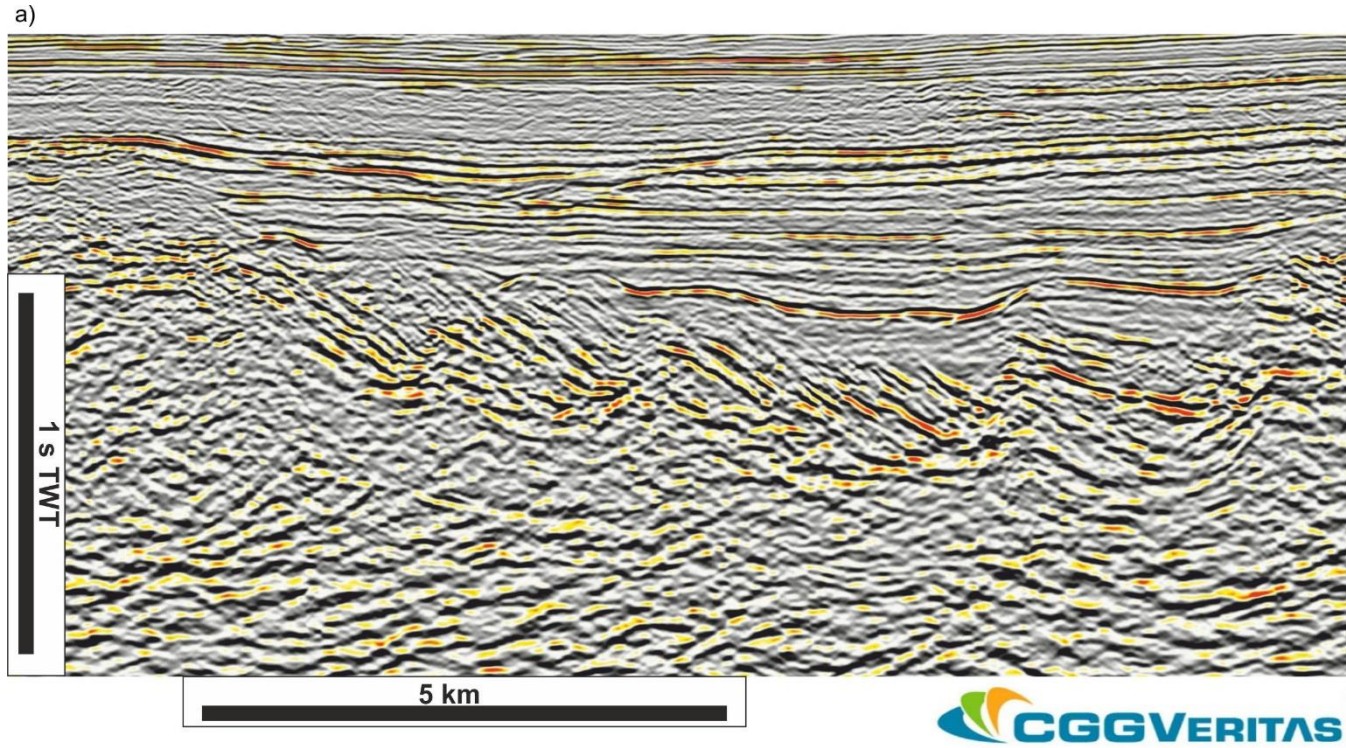

b)

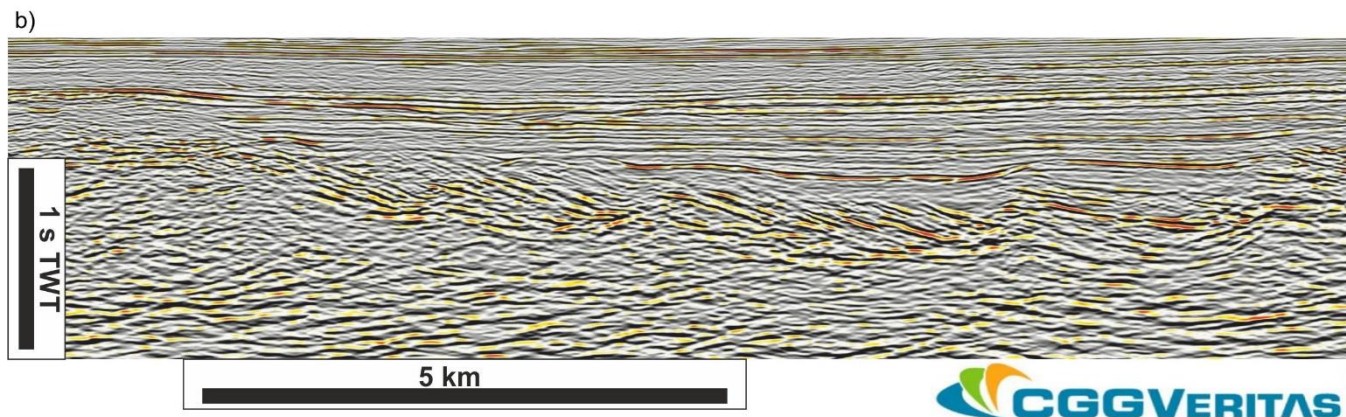

**Figure 2**

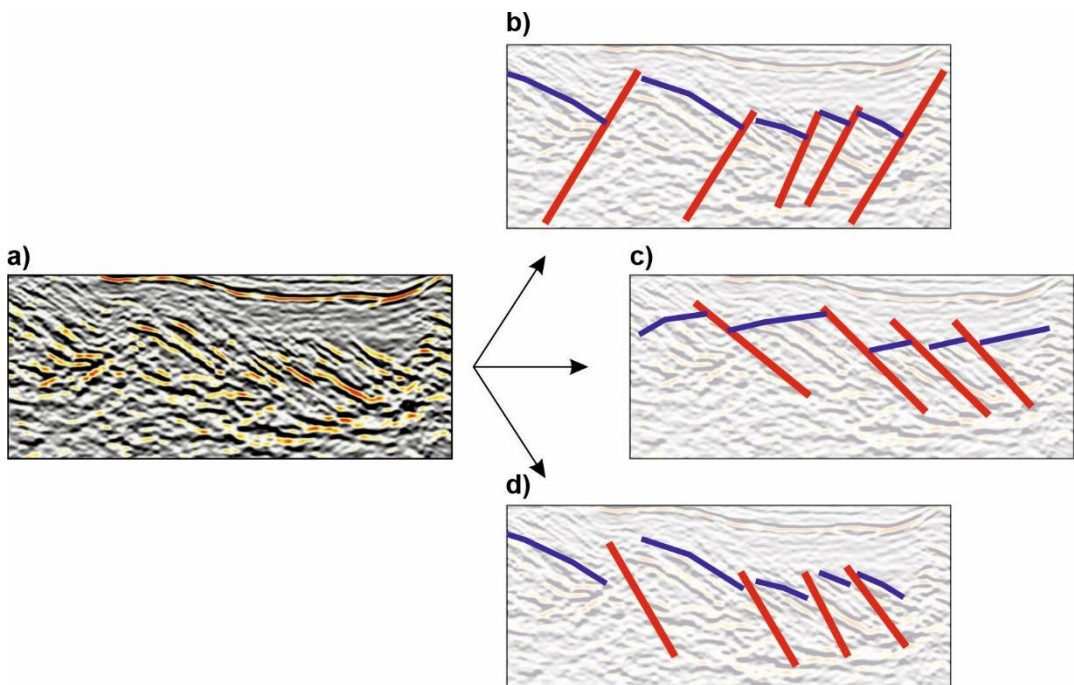

**Figure 3**

| | Aggregated Total | VE = 1:4<br>(88 interpretations – 55%) | VE = 1:2<br>(72 interpretations – 45%) |
|---|---|---|---|
| **Fault direction**<br><br>L – left<br>R – right | L (52)<br>44%<br>56%<br>R (67) | L (27)<br>41%<br>59%<br>R (39) | L (25)<br>47%<br>53%<br>R (28) |
| **Fault type**<br><br>R – reverse<br>N – normal | R (32)<br>32%<br>68%<br>N (67) | R (17)<br>32%<br>68%<br>N (36) | R (15)<br>33%<br>67%<br>N (31) |

**Figure 4**

| | Fault motion R – reverse N – normal | VE = 1:4 + Fault motion | VE = 1:2 + Fault motion |
|---|---|---|---|
| **LEFT dipping** | R (1) 3% 97% N (35) | 4R (1) 5% 95% 4N (19) | 2R (0) 100% 2N (16) |
| **RIGHT dipping** | N (24) 44% 56% R (31) | 4N (9) 39% 61% 4R (14) | 2N (15) 47% 53% 2R (17) |

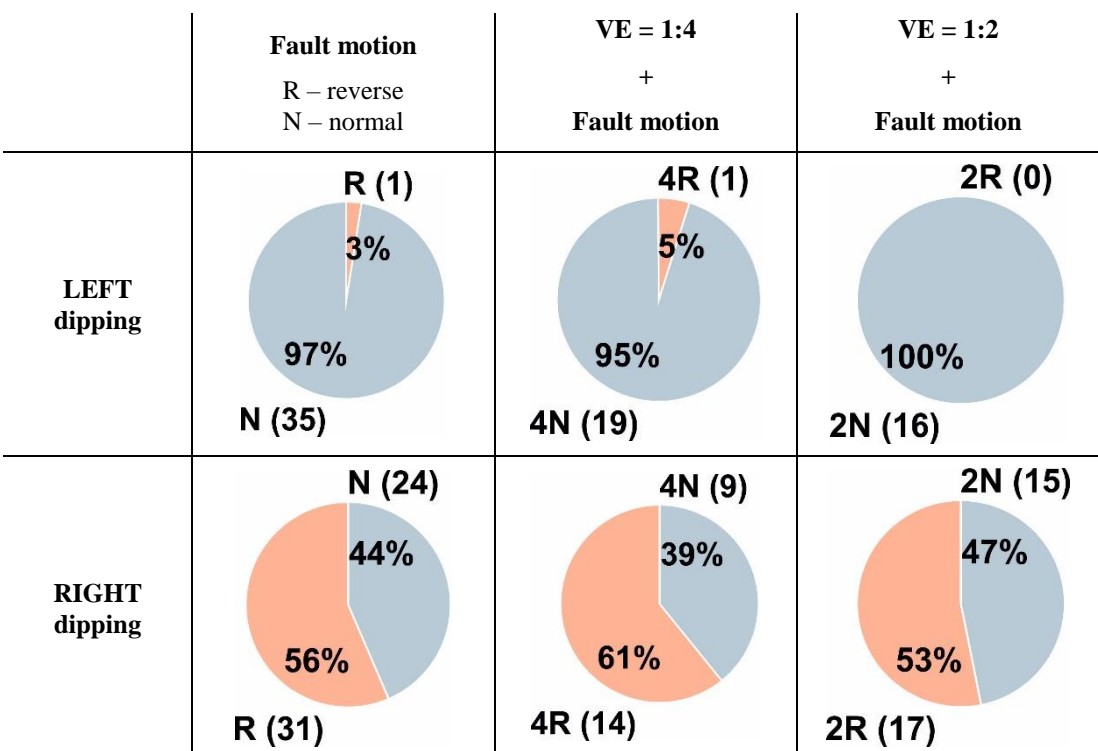

**Figure 5**

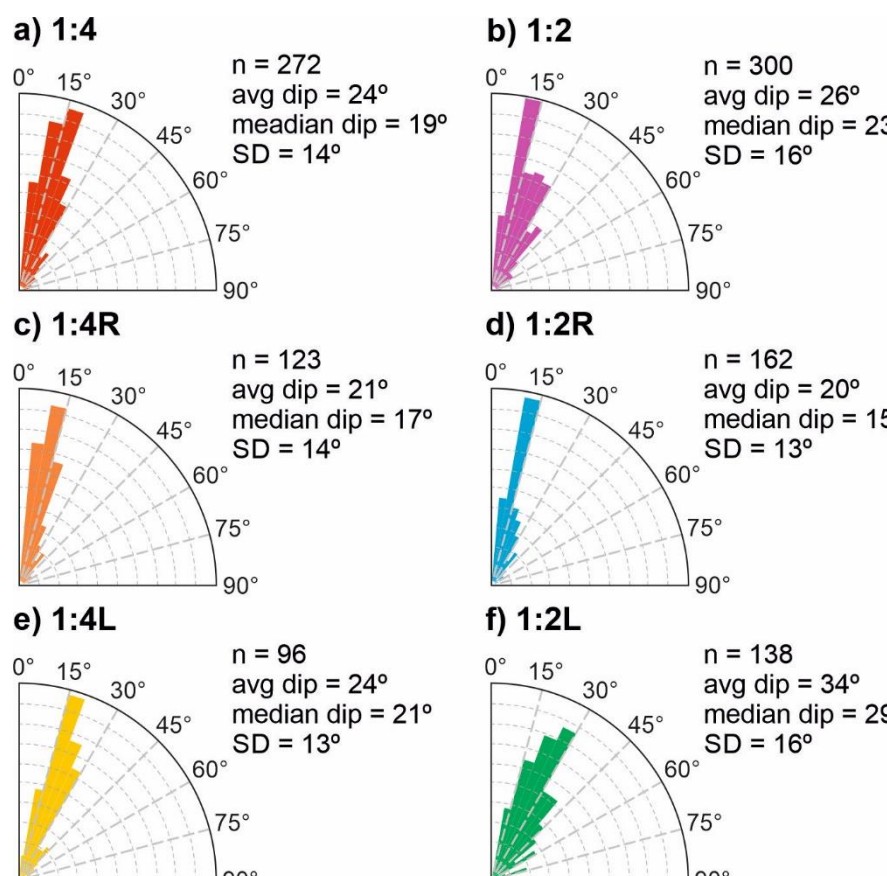

**a) 1:4**
0° 15° 30° 45° 60° 75° 90°
n = 272
avg dip = 24°
meadian dip = 19°
SD = 14°

**b) 1:2**
0° 15° 30° 45° 60° 75° 90°
n = 300
avg dip = 26°
median dip = 23°
SD = 16°

**c) 1:4R**
0° 15° 30° 45° 60° 75° 90°
n = 123
avg dip = 21°
median dip = 17°
SD = 14°

**d) 1:2R**
0° 15° 30° 45° 60° 75° 90°
n = 162
avg dip = 20°
median dip = 15°
SD = 13°

**e) 1:4L**
0° 15° 30° 45° 60° 75° 90°
n = 96
avg dip = 24°
median dip = 21°
SD = 13°

**f) 1:2L**
0° 15° 30° 45° 60° 75° 90°
n = 138
avg dip = 34°
median dip = 29°
SD = 16°

**Figure 6**

**a) 1:4LN**

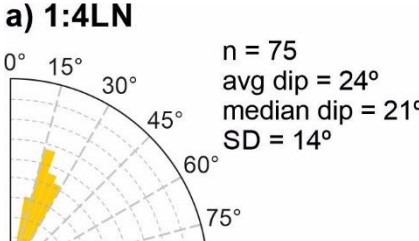

n = 75
avg dip = 24°
median dip = 21°
SD = 14°

**b) 1:2LN**

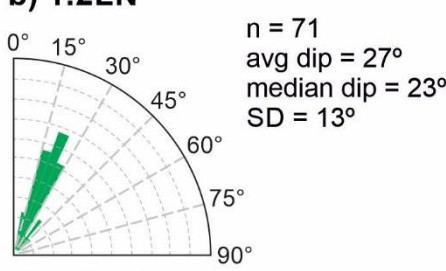

n = 71
avg dip = 27°
median dip = 23°
SD = 13°

**c) 1:4RR**

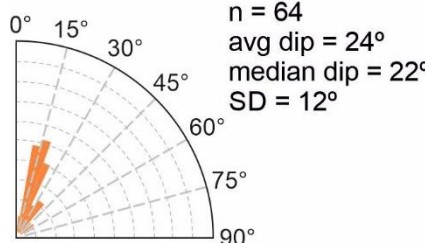

n = 64
avg dip = 24°
median dip = 22°
SD = 12°

**d) 1:2RR**

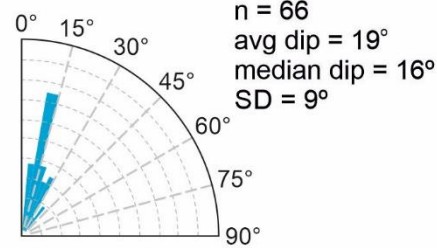

n = 66
avg dip = 19°
median dip = 16°
SD = 9°

**e) 1:4RN**

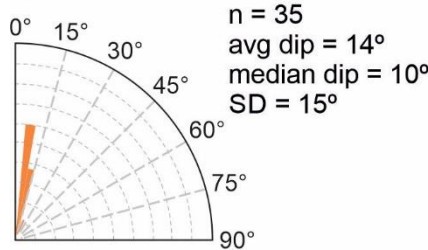

n = 35
avg dip = 14°
median dip = 10°
SD = 15°

**f) 1:2RN**

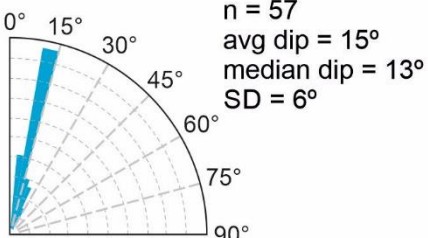

n = 57
avg dip = 15°
median dip = 13°
SD = 6°

**Figure 7**

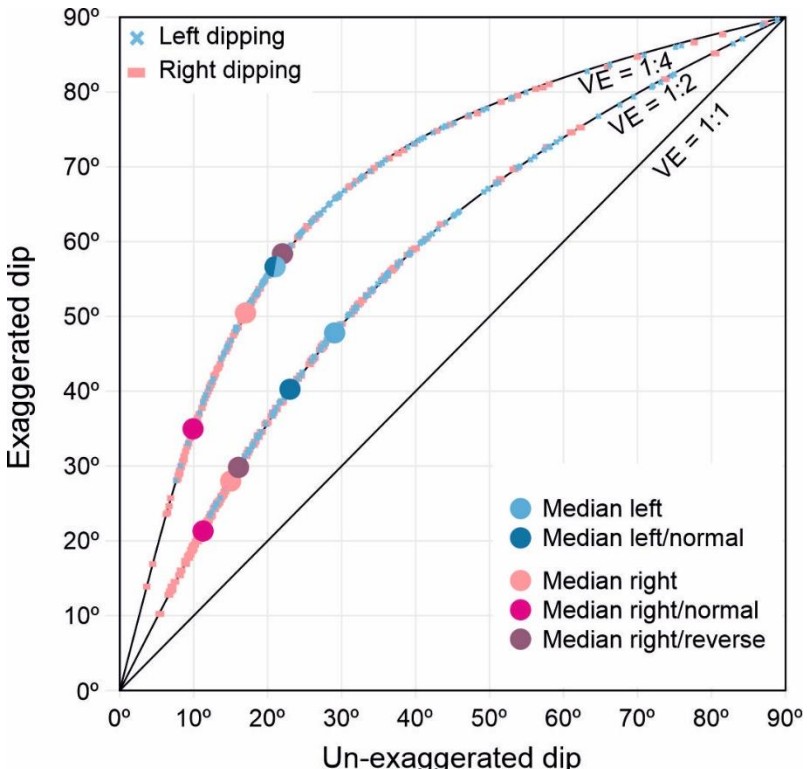

**Figure 8**

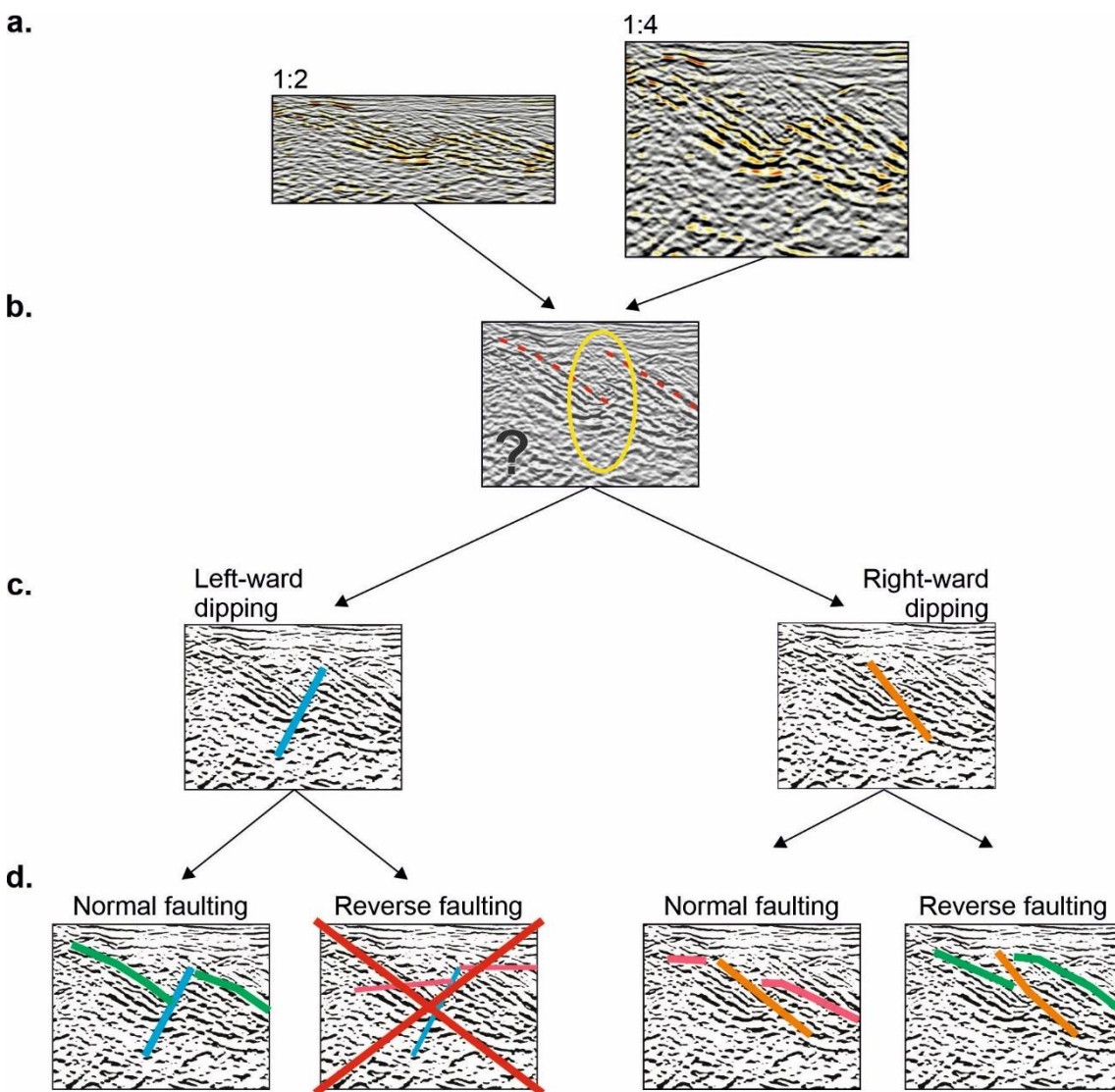

**Figure 9**