# Peer review of "Fault interpretation in seismic reflection data: an experiment analysing the impact of conceptual model anchoring and vertical exaggeration"

_Solid Earth, 2019_

## Referee Comment (RC1) · Charlotte Botter (Referee) · 5 Jun 2019

Charlotte Botter – June 2019

This manuscript addresses a very interesting and useful problem, the bias we, as geologists, seismic interpreters or geophysicists, have when interpreting seismic data. This study has even more impact as no context at all was given to the interpreters and they were limited in time for their interpretation. The results highlight quite a variability in the finale interpretation with a majority of the students falling into the most common (easiest?) interpretation (extensional settings, with faults dipping towards the right). Does it reflect the background and familiarity of the students towards extensional over

compressional tectonic settings? The title and the abstract are promising, however the content of the paper lacks of clarity, clear objectives and outcomes, and conclusions that support the initial statements. The two terms that require a clear definition are conceptual model and uncertainty (I think you are mixing uncertainty vs variability in the paper).

I recommend a better organisation of the manuscript and its content, and a title (the second part of the title 'evidence of conceptual model uncertainty and anchoring' is misleading in view of the results/conclusions) that reflects better the content before publication in Solid Earth.I have ticked major revisions but this could be considered as minor as it is mainly regarding the writing and presentation of the results rather than the scientific approach (I do believe that this type of study is very important and exciting!).

I have attached the manuscript pdf with specific comments at various locations of the text and Figures but you can find more general comments and questions below. The text is well written overall and I did not spot spelling mistake but highlighted some inconsistencies between Figures annotations, captions and text.

1. I think the paper requires a clear and exhaustive definition of a conceptual model and what goes into it. Then you can refine the possible and plausible conceptual models that are appropriate to this case study. By reading the manuscript I often got the impression that the conceptual model was the result of the final interpreted section but at the same time that the conceptual model influenced the interpretation. It is not clear what is the relationship between the conceptual model and the interpretation: - Does the conceptual model influence the interpretation (c. model => interpretation)? - Does the interpretation become the conceptual model (interpretation => c. model)? - Are the conceptual model and interpretation developed at the same time/simultaneously (c. model <=> interpretation)? This becomes quite important when you talk about uncertainty and anchoring of the conceptual model.

2. The title states "evidence of conceptual model uncertainty", however uncertainties

are not clearly defined, addressed or dealt with within the manuscript. If there is no clear definition of uncertainty (qualitative and/or quantitative) and paragraph/section in the Discussion, this should be removed from the title. When at the end of the conclusions, you say "uncertain geological and geophysical data", it seems that uncertain here only means that there is no unique interpretation. This case your study only illustrates that that there is a range of different yet plausible interpretations for one given seismic line (which was provided with no colour scale, which is supposed to be as standard information than a scale bar). It doesn't really highlight any clear uncertainty (why these different interpretations, are they related to the background, to the colour scale, or else? As you state that vertical exaggeration does not have a real impact).

3. I am a bit dubious on the fact that this study proves/shows that the conceptual model is anchored. Given the short time students were left with for the interpretation (15-30 min), it seems that they would have not been able to supply a different conceptual model to the one they started with. Indeed, if the students were provided with additional data or context after a first interpretation, would they update their model or will they keep it unchanged? Is it possible to define 'anchored' conceptual models without taking that into account? If the authors are satisfied with this simplified definition of anchoring, they should discuss it or at least make it clearer in the manuscript.

4. I understand that this study is quite exciting but it would even be more if it were directly related to the background of the interpreter. In the appendix you provide the survey and questions handed to the students. A summary of the results of this survey should also be added to the paper/appendix to know if Normal vs Reverse fault is falling mostly for people that interpret often or with no knowledge about seismic interpretation. I think this survey is also part of the bias that forms the conceptual model.

5. One additional figure summarising clearly the work, such as the different interpretations, conceptual models and implications (such as which interpretation is the most probable) is necessary to fully comprehend the implications of this work.

Please also note the supplement to this comment:
https://www.solid-earth-discuss.net/se-2019-66/se-2019-66-RC1-supplement.pdf

[Figure]

**Supplement:**

[revised manuscript text omitted]

a)

[Figure]

b)

[Figure]

Figure 2

[Figure]

[Figure]

[Figure]

**Figure 3**

[Figure]

[Figure]

| | Aggregated Total | VE = 1:4 (88 interpretations – 55%) | VE = 1:2 (72 interpretations – 45%) |
|---|---|---|---|
| **Fault direction** L – left R – right | L (52) 44% 56% R (67) | L (27) 41% 59% R (39) | L (25) 47% 53% R (28) |
| **Fault type** R – reverse N – normal | R (32) 32% 68% N (67) | R (17) 32% 68% N (36) | R (15) 33% 67% N (31) |

**Figure 4**

[Figure]

[Figure]

**Figure 5**

[Figure]

[Figure]

[Figure]

**Figure 6**

[Figure]

[Figure]

**Figure 7**

[Figure]

[Figure]

[Figure]

**Figure 8**

---

## Referee Comment (RC2) · Javier Tamara (Referee) · 5 Jun 2019

Summary

The paper address the impact of vertical exaggeration on the interpretation of seismic sections. Seismic sections in time with two different vertical scales (1:2 and 1:4) were given to students for interpretation. This resulted in 3 main interpretations of structural styles seen in both scales. The authors measured fault dips of individual interpretations and analysed the results of all the data.

General comments

1. Effect of background and expertise

In the introduction, it is mentioned that previous studies have shown that the background of the interpreters is essential. However, the authors do not seem to take into account the workflows and methodologies used during the interpretation, which may be related to the knowledge and experience of the interpreter. I suggest that the result of the study should be filtered based on knowledge and experience. You can divide the result, e.g. as undergraduate vs postgraduate, and use additional information such as knowledge and attendance to courses in structural geology or seismic interpretation to rank the knowledge of the interpreter. That information is already in the data collected.

2. Anchoring and bias effect

The authors interpreted that defining a reflection or set of reflections as horizons or faults may represent a form of anchoring. However, this seems different from the concept and examples described in the introduction where new data is given after an initial interpretation, and the interpreter does not see the necessity to adjust their interpretation.

In the discussion of anchoring, the authors mentioned "horizons cutting reflections" as an element that suggest anchoring. However, it is not possible to know how and where in the seismic section the interpreters defined the horizons and the faults, or where did they start the interpretation. Moreover, the authors also discussed that they could not know if the students changed their minds during the interpretation, or what elements within the seismic section were considered in the process. Therefore, it is not clear how the anchoring bias was defined.

The fact that experience was not taken into account is also a problem. "horizons cutting reflections" may reflect a lack of understanding of seismic interpretation.

3. Vertical exaggeration analysis

Although the exercise is related to the perception of the interpreter to the scale, the

authors based their analysis on the interpretation of seismic sections in time. It seems that the authors consider 1:2 and 1:4 represent vertically exaggerated displays of a 1:1 section. These scales are more likely to be display factors for seismic sections in time. A 1:1 display factor in time will be significantly different from a 1:1 section displayed in depth. The same problem will apply for other display factors in time, which will not be representative of the real scale of the structures in depth.

Moreover, the depth section will depend on the velocity model used for depth-conversion. For example, assuming an unrealistic and simple model of a constant velocity of 5000 m/s throughout the section, the 1:2 display ratio you used will be equivalent to a 1:∼0.7 depth ratio, whereas at a velocity of 4000m/s, this will be equal to a 1:∼0.9 depth ratio. This means the section with 1:2 display is more likely horizontally stretched and not vertically exaggerated. The analysis of the impact of vertical exaggeration should, therefore, be performed in a depth section and possible in PSDM section.

I suggest you depth-convert a section, so you have a reference for what a 1:1 section in depth looks like. Then compare these to the time sections given to the students and analyse the result taking into account the depth section. You can also consider additional display factors (e.g. 1:6, 1:8) in time to complement your study.

4. Fault dip data

The author should clearly state in the text that the measurements were made for comparison and are not representative of real fault dips.

The analyses of fault dips should be divided based on the conceptual models, as different assumptions were made for these. Mixing data from different conceptual models based on dip direction as in figures 4 & 6 seems inadequate. The authors mentioned that right-dipping reverse faults required higher angles than right-dipping normal faults so that both populations will differ due to the assumption made during the interpretation. Hence, these should be treated separately as in figure 5 & 7.

The data shows significant variability suggesting that the average should not be used.

In figure 8, the analysis of fault dips should take into account the position of the faults within the section. As they are, the results show too much dispersion and mix different conceptual models (right-dipping faults). It is not possible to correlate faults between different display scales. Hence, it is not possible to know if the perception changed due to the scale. As currently displayed (rose diagrams (Figure 7) and curves (Figure 8)), the results are difficult to interpret and should not be used.

I suggest that the authors subdivide the dataset based on the conceptual models and use plots of horizontal distance vs fault dip. For example, you can compare the horizontal distribution of the interpreted faults to see which faults are recurrent in the interpretations, and are located in similar places. Then for each location where faults are repeated, you can analyse the "dip" distribution and calculate a representative value for that population. This can then be compared between scales and plotted in figure 8.

Final remarks

Vertical exaggeration and anchoring are important aspects that should be taken into account during seismic interpretation. Research on the impact of these in the outcome of the interpretations can contribute to the seismic interpretation workflow. However, the way the experiments are designed and the way results are present are crucial.

Although the authors try to discuss the importance of vertical exaggeration and anchoring, the paper in its current state does not give support for their conclusions. The effect of anchoring is based on assumptions, and the existence of bias cannot be evaluated from the data and the analyses presented. The experiment, as described, is unlikely to support a discussion on anchoring and bias.

The use of time sections makes the results uncertain, and their analyses do not include the background and experience of the interpreters.

I suggest the methodology used to analyse the data needs to be modified. Current diagrams are difficult to analyse and sometimes combine the result of different conceptual models. Some of these do not support their discussion points.

I suggest the author should revisit the methodology and parameters used to analyse the data, as well as the way the results are displayed. This inevitably requires major changes in the way the paper is presented, including significant modifications to the text and figures. The discussion and conclusions should be reconsidered after these modifications.

Please also note the supplement to this comment:
https://www.solid-earth-discuss.net/se-2019-66/se-2019-66-RC2-supplement.pdf

[Figure]

**Supplement:**

[revised manuscript text omitted]

Figure 1

[Figure]

a)

[Figure]

b)

[Figure]

**Figure 2**

[Figure]

[Figure]

[Figure]

**Figure 3**

[Figure]

[Figure]

|  | **Aggregated Total** | **VE = 1:4** (88 interpretations – 55%) | **VE = 1:2** (72 interpretations – 45%) |
|---|---|---|---|
| **Fault direction**
L – left
R – right | L (52) 44% / 56% R (67) | L (27) 41% / 59% R (39) | L (25) 47% / 53% R (28) |
| **Fault type**
R – reverse
N – normal | R (32) 32% / 68% N (67) | R (17) 32% / 68% N (36) | R (15) 33% / 67% N (31) |

**Figure 4**

[Figure]

[Figure]

**Figure 5**

[Figure]

[Figure]

[Figure]

**Figure 6**

[Figure]

[Figure]

[Figure]

**Figure 7**

[Figure]

[Figure]

[Figure]

**Figure 8**

---

## Author Comment (AC1) · 19 Aug 2019

**Responses to Reviewer #1**

*This manuscript addresses a very interesting and useful problem, the bias we, as geologists, seismic interpreters or geophysicists, have when interpreting seismic data. This study has even more impact as no context at all was given to the interpreters and they were limited in time for their interpretation. The results highlight quite a variability in the finale interpretation with a majority of the students falling into the most common (easiest?) interpretation (extensional settings, with faults dipping towards the right). Does it reflect the background and familiarity of the students towards extensional over compressional tectonic settings? The title and the abstract are promising, however the content of the paper lacks of clarity, clear objectives and outcomes, and conclusions that support the initial statements. The two terms that require a clear definition are conceptual model and uncertainty (I think you are mixing uncertainty vs variability in the paper).*

*I recommend a better organisation of the manuscript and its content, and a title (the second part of the title 'evidence of conceptual model uncertainty and anchoring' is misleading in view of the results/conclusions) that reflects better the content before publication in Solid Earth. I have ticked major revisions but this could be considered as minor as it is mainly regarding the writing and presentation of the results rather than the scientific approach (I do believe that this type of study is very important and exciting!).*

Please find attached the manuscript with modifications added to reflect reviewers' comments and concerns. Unless specifically mentioned below, we have introduced all changes suggested in the supplementary by the reviewer.

*1. I think the paper requires a clear and exhaustive definition of a conceptual model and what goes into it. Then you can refine the possible and plausible conceptual models that are appropriate to this case study. By reading the manuscript I often got the impression that the conceptual model was the result of the final interpreted section but at the same time that the conceptual model influenced the interpretation. It is not clear what is the relationship between the conceptual model and the interpretation: - Does the conceptual model influence the interpretation (c. model => interpretation)? - Does the interpretation become the conceptual model (interpretation => c. model)? - Are the conceptual model and interpretation developed at the same time/simultaneously (c. model <=> interpretation)? This becomes quite important when you talk about uncertainty and anchoring of the conceptual model.*

We have addressed this comment by adding a clear definition of conceptual model in the introduction as suggested by Reviewer #1, and located it upfront in the second paragraph of the introduction:

*Geoscientists employ mental models (or "conceptual models") that incorporate their observations and that conform to their understanding of the world (Shipley and Tikoff, 2016). These conceptual models are dynamically modified or renewed with the arrival of new observations (input information), and are used to produce predictions (inferences) that can help to answer questions about the world (Shipley and Tikoff, 2017). When confronted with geological data, interpreters need to apply different conceptual models, acquired during their training and past experience, together with robust interpretation methodologies, in order to produce interpretations that honour the data, particularly in areas of great uncertainty (Bond et al., 2007; Bond et al., 2015). Interpreters need to be able to identify key elements (e.g. growth geometries, regional level) and employ different validation techniques (e.g. restoration, attribute analyses) that allow differentiation between (a priori similar) conceptual models (Bond, 2015). Conceptual models are therefore the basis of the interpretation, as they provide the necessary criteria to make sense of the data (Frodeman, 1995).*

We have then better linked this to the following section on anchoring bias. New text added in italics:

To deal with uncertainty, interpreters employ heuristics (or 'rules of thumb') in the process of generating the conceptual models, and that makes them subject to a broad range of cognitive biases (Kahneman et al., 1982). One of these biases is related to the capability of interpreters to adjust their interpretations from their initial ideas or conceptual models. This type of bias, called anchoring, has been identified in many decision-making processes since it was first described by Tversky and Kahneman (1974), and takes place in the seismic interpretation process. Rankey and Mitchell (2003) investigated the effect of anchoring in an interpretation experiment by asking interpreters to reassess their seismic interpretations after being provided with additional well data. Their work shows that most interpreters did not feel that their interpretations needed to change substantially, in spite of data showing changes in porosity and net-to-gross predictions that did not fit with their initial interpretations. Their results suggest that interpreters were anchored to their initial conceptual models, and that they were reluctant to change their mind in

light of new data. In a different experiment, Bond et al. (2007) observed that participants asked for the geographical location of the section and suggested that interpreters could use this information to build their conceptual models, by using geographically specific knowledge of e.g. the relevant tectonic setting to anchor their interpretation. For example, an interpreter knowing a seismic section was from the North Sea may assume a conceptual model based on an extensional tectonic regime and will consciously and unconsciously look for normal faults in the seismic data. However, if the conceptual model is wrong, e.g. there is significant inversion in the seismic section, the interpretation could be compromised. *So although conceptual models can be dynamically modified or renewed with the arrival of new observations, as described by Shipley and Tikoff (2017) and others, anchoring bias often results in limited adjustment from initial models.* Thus, although conceptual models are needed to develop geologically sound interpretations, they can also create anchors to potentially erroneous outcomes.

2. The title states "evidence of conceptual model uncertainty", however uncertainties are not clearly defined, addressed or dealt with within the manuscript. If there is no clear definition of uncertainty (qualitative and/or quantitative) and paragraph/section in the Discussion, this should be removed from the title. When at the end of the conclusions, you say "uncertain geological and geophysical data", it seems that uncertain here only means that there is no unique interpretation. This case your study only illustrates that that there is a range of different yet plausible interpretations for one given seismic line (which was provided with no colour scale, which is supposed to be as standard information than a scale bar). It doesn't really highlight any clear uncertainty (why these different interpretations, are they related to the background, to the colour scale, or else? As you state that vertical exaggeration does not have a real impact).

We have changed the title of the manuscript to better reflect the findings presented throughout the text, and revised the presentation of the objectives of the study in the introduction and their interrelation in the discussion and conclusions sections. The new title is:

*Evidence of anchoring to initial conceptual models during interpretation of a vertically exaggerated seismic section*

Regarding uncertainty vs variability, we use the variability or range in interpretations as indicative of the range of interpretational uncertainty (see also Schaaf et al. 2019 (this volume)). In doing so we have considered what influences the variability in the interpretations proposing that anchoring to initial conceptual models appears to influence the range in interpretations. We also considered whether vertical exaggeration can introduce a greater range in interpretations and hence interpretational uncertainty. However, we found that vertical exaggeration had a subdued influence in the interpretation compared with anchoring to conceptual models. We have made this clearer in the updated introduction and the discussion/conclusions and recommendations sections.

3. I am a bit dubious on the fact that this study proves/shows that the conceptual model is anchored. Given the short time students were left with for the interpretation (15-30 min), it seems that they would have not been able to supply a different conceptual model to the one they started with. Indeed, if the students were provided with additional data or context after a first interpretation, would they update their model or will they keep it unchanged? Is it possible to define 'anchored' conceptual models without taking that into account? If the authors are satisfied with this simplified definition of anchoring, they should discuss it or at least make it clearer in the manuscript.

The reviewer is correct in that we surmise the anchoring from the interpretation process we asked the participants to undertake and the outcome of that interpretation process rather than through provision of additional data. In the original Tversky and Khaneman experiment anchoring was demonstrated by providing an initial value from which the participants were then asked to give an estimate (they were not provided with additional information). In contrast interpreters in the experiment by Rankey and Mitchell (2003) were given additional information and showed that interpreters were reluctant to adapt their interpretations to new information. We suggest that the final interpretation outcome in our experiment results from participants initial fault feature selection (i.e. right or left dipping elements in the seismic image data). In this way their initial conceptual model and its application provides the anchor, in much the same way as the initial values given by Tversky and Khaneman in their experiment provide the anchor to future value estimates. We described this in the third paragraph of the discussion:

In summary, from the analysis of the fault and horizon interpretations of participants, three conceptual models are identified (Figure 3) that have been applied in interpretations of the data. What we do not know is how the individual participants honed onto their 'chosen' conceptual model. The participants were prompted to interpret the faults as their main task in the experiment instructions, and as a secondary element to interpret a horizon to show fault motion; *an interpretation sequence as shown in figure 9*. We should state that we cannot be sure that all participants followed this workflow, but we have no evidence to suggest that they did not.

Irrespective of the exact interpretation sequence, we suggest that once participants started interpreting certain 'features' in the reflection seismic image data as faults or horizons, they became anchored to an initial conceptual model and fitted the rest of their interpretation to this model. *Consequently, we suggest that interpreters were likely anchored to their initial thoughts on the direction of dip of the faults and the rest of their interpretation is determined by this initial fault model, irrespective of whether later interpretative elements conform to the data e.g. horizons cutting reflections, as seen in Figure 3, this has previously been reported by Rankey and Mitchell (2003) and Torvela and Bond (2011). Although, there appears to be a threshold of tolerance for data disconfirmation. Note that no left-ward dipping faults with a reverse sense of motion have been interpreted, in which horizons would very distinctively have cut seismic reflectors (see figure 9d).*

*Experience and knowledge is expected to have played a key role in informing the initial observations that led to selection of a conceptual model at the beginning of the interpretation. We purposely chose a student only cohort to mitigate against the competing effects of experience and knowledge with other factors we wanted to test. To ensure this was the case we have analysed the data for differences in interpretation outcome between students from different Universities and between undergraduate and postgraduate students. This analysis shows no strong evidence that experience had an effect on interpretation outcome. We therefore interpret our data as showing that although initial interpretations are informed by the data, these first conceptual models become anchored to and are applied irrespective of whether they later conform to all the data, albeit to a threshold. This suggests that initial conceptual models play a dominant role in interpretation outcome.*

4. I understand that this study is quite exciting but it would even be more if it were directly related to the background of the interpreter. In the appendix you provide the survey and questions handed to the students. A summary of the results of this survey should also be added to the paper/appendix to know if Normal vs Reverse fault is falling mostly for people that interpret often or with no knowledge about seismic interpretation. I think this survey is also part of the bias that forms the conceptual model.

This comment was also raised by Reviewer #2 and a detailed response can be found in the responses to Reviewer #2. In summary, we did a broad assessment of the effect of experience in the interpretation results and did not find any conclusive correlation. There is a disparity in the number of undergraduate vs postgraduate participants (122 vs 34 participants, respectively), and we do not feel their experience levels are potentially so dissimilar to treat the dataset as two different cohorts. We nevertheless found a small difference in the fault types interpreted by the two cohorts, and therefore added a paragraph to the text to state that further research in this matter is needed:

*There are minor differences between the fault type interpreted by undergraduate vs postgraduate students, but the disparity in the size of the two cohorts (122 vs 34 interpretations, respectively) does not allow us to pursue this line of research. The effect of level of education and experience in seismic interpretation has been raised in the past (e.g. Bond et al., 2012; Alcalde et al., 2017b), so further work in this area could provide fruitful in better understanding interpretation processes.*

5. One additional figure summarising clearly the work, such as the different interpretations, conceptual models and implications (such as which interpretation is the most probable) is necessary to fully comprehend the implications of this work.

We have added a new figure (Figure 9) to summarise the proposed interpretation workflow. In this figure we propose that, independent of the vertical exaggeration of the seismic section interpreted (i.e. 1:2 or 1:4 vertical exaggeration), participants interpreted the faults first, as requested, and the rest of the interpretation was anchored by this initial fault selection.

*Figure 9: Proposed interpretation sequence. (a) The seismic images were presented in both 1:2 and 1:4 vertical exaggerations. (b) Independently of the image interpreted, the participants of the experiment faced the problem of how to interpret the right-ward dipping structures and the chaotic seismic fabric. (c) Participants interpreted the central fabric either as a left-ward (blue) or right-ward (orange) dipping fault, which consequently triggered (d) the horizon interpretation determining the motion (normal, green horizons; and reverse, pink horizons) of the fault. The left-ward dipping, reverse faulting interpretation (crossed out in red) is too difficult to fit to the seismic data, and so only one participant chose this interpretation.*

**a.**

1:2

1:4

**b.**

?

**c.**

Left-ward
dipping

Right-ward
dipping

**d.**

Normal faulting    Reverse faulting    Normal faulting    Reverse faulting

---

## Author Comment (AC2) · 19 Aug 2019

**Responses to Reviewer #2**

General comments

1. Effect of background and expertise. In the introduction, it is mentioned that previous studies have shown that the background of the interpreters is essential. However, the authors do not seem to take into account the workflows and methodologies used during the interpretation, which may be related to the knowledge and experience of the interpreter. I suggest that the result of the study should be filtered based on knowledge and experience. You can divide the result, e.g. as undergraduate vs postgraduate, and use additional information such as knowledge and attendance to courses in structural geology or seismic interpretation to rank the knowledge of the interpreter. That information is already in the data collected.

As suggested by Reviewer #2, we have conducted a broad assessment of the effect of experience of the participants in the interpretation results. A summary of this assessment is summarised in the table below:

Table 1: Statistics for the interpreted fault directions (left 'L' or right 'R'), and motions (normal 'N' or reverse 'R'), of the total interpretations (left, in blue) and separated by education of the participants (undergraduate, centre, in orange, or postgraduate students, right, in green).

| Total interpretations | | | | | Undergraduate | | | | | Postgraduate | | | |
|---|---|---|---|---|---|---|---|---|---|---|---|---|---|
| Direction | | Type | | | Direction | | Type | | | Direction | | Type | |
| L | R | N | R | | L | R | N | R | | L | R | N | R |
| 52 | 67 | 67 | 32 | | 39 | 50 | 52 | 23 | | 13 | 14 | 13 | 9 |
| 43.7% | 56.3% | 67.7% | 32.3% | | 43.8% | 56.2% | 69.3% | 30.7% | | 48.1% | 51.9% | 59.1% | 40.9% |

An issue encountered in considering the data in 'experience' cohorts is that there are a different number of participants in each cohort, with 122 undergraduate students and only 34 postgraduate students. Nonetheless, separating the results into experience cohorts shows little difference in the overall ratios of the types of fault interpreted. The general trend observed in the total interpretations (i.e. greater percentage of right-dipping faults and normal fault types) are conserved.

In this experiment, the original hypothesis was that vertical exaggeration could have a strong effect in interpretation. However, our results turned out to show that conceptual models might play a stronger role than perceptual biases such as changes in vertical exaggeration. Experience, as highlighted by Reviewer #2 and by studies referenced in the manuscript introduction (e.g. Bond et al., 2012, *Geology* or Alcalde et al., 2017b, *Journal of Structural Geology*), can impact interpretation results, but we do not see this here.

Nevertheless, we have added a paragraph to the results section to describe this issue and to encourage the study of this effect in future works. If the editor would like us to add the above tables into a data repository, to support our findings, we are willing to do so:

*To check if other factors, specifically: educational background and experience, were influencing interpretation outcome we also analysed the data for disparities between different University cohorts and between undergraduate and postgraduate students.* There are no major differences in the analysed results across student cohorts from different Universities, *or between undergraduate and postgraduate students. For the latter cohort the difference in numbers (undergraduate (126) vs postgraduate (35) students) is large and does not allow easy comparison; despite this the ratios of leftward and rightward dipping faults and the sense of off-set is consistent across the cohorts. The effect of level of education and experience in seismic interpretation has been raised in the past (e.g. Bond et al., 2012; Alcalde et al., 2017b) and we suggest that this is still an area of interest for future work.*

Our analysis of experience within the cohort has been taken further in this revision following a comment made by Reviewer #2 in the comments in the manuscript Section 4.1: Can these result from the lack of experience of the interpreters? How does this impact your results?

We analysed the full conceptual models by experience level (undergraduate vs postgraduate), and did not find any remarkable difference between the models interpreted and the level of studies, given the number of participants per cohort:

| | Total interpretations (with sense of fault motion)* | | Right-ward dipping, normal faults | | Right-ward dipping, thrust faults | | Left-ward dipping, normal faults | |
|---|---|---|---|---|---|---|---|---|
| Undergraduate | 68 | 76% | 19 | 28% | 22 | 32% | 27 | 40% |
| Postgraduate | 21 | 24% | 4 | 19% | 9 | 43% | 8 | 38% |

*Note that the number of interpretations do not add up to 161 when summing the subcohort results: this is because not all the interpreters featured a clear interpretation model (i.e. the motion of the faults was not clear).

2. Anchoring and bias effect. The authors interpreted that defining a reflection or set of reflections as horizons or faults may represent a form of anchoring. However, this seems different from the concept and examples described in the introduction where new data is given after an initial interpretation, and the interpreter does not see the necessity to adjust their interpretation. In the discussion of anchoring, the authors mentioned "horizons cutting reflections" as an element that suggest anchoring. However, it is not possible to know how and where in the seismic section the interpreters defined the horizons and the faults, or where did they start the interpretation. Moreover, the authors also discussed that they could not know if the students changed their minds during the interpretation, or what elements within the seismic section were considered in the process. Therefore, it is not clear how the anchoring bias was defined. The fact that experience was not taken into account is also a problem. "horizons cutting reflections" may reflect a lack of understanding of seismic interpretation.

See comments on anchoring made to reviewer 1, copied here for clarity:

The reviewer is correct in that we surmise the anchoring from the interpretation process we asked the participants to undertake and the outcome of that interpretation process rather than through provision of additional data. In the original Tversky and Khaneman experiment anchoring was demonstrated by providing an initial value from which the participants were then asked to give an estimate (they were not provided with additional information). In contrast interpreters in the experiment by Rankey and Mitchell (2003) were given additional information and showed that interpreters were reluctant to adapt their interpretations to new information. We suggest that the final interpretation outcome in our experiment results from participants initial fault feature selection (i.e. right or left dipping elements in the seismic image data). In this way their initial conceptual model and its application provides the anchor, in much the same way as the initial values given by Tversky and Khaneman in their experiment provide the anchor to future value estimates. We described this in the third paragraph of the discussion:

On the interpretation process – we asked participants to interpret the faults first and then a horizon to show fault off-set. Although we cannot be sure they followed this process, the final interpretation outcomes (multiple faults interpreted and a single horizon) leads us to believe that the participants followed the workflow as instructed. We make this clear in the manuscript, section 2. Experimental set up, lines 23-25:

' The participants were asked to "interpret the main faults crossing the section as deep as possible", as well as to add a "sedimentary horizon to mark the displacement",..'

In the discussion on anchoring we discuss the potential for participants not to have followed the workflow requested. We felt that it was important to raise this as a potential weakness in our methodology, but as stated do not feel it impacts our findings. We have updated this sentence to reflect that (page 8, lines 8-9):

*'We should state that we cannot be sure that all participant followed this workflow, but we have no evidence to suggest that they did not.'*

On how we define conceptual models and anchoring – We have added in a new section to the introduction that discusses previous work and how this relates to our study ( pasted below – new text in *italics*):

*Geoscientists employ mental models (or "conceptual models") that integrate their observations and that conform their understanding of the world (Shipley and Tikoff, 2016). When confronted with geological data, interpreters need to apply different conceptual models, acquired during their training and past experience (through learning), together with robust interpretation methodologies, in order to produce interpretations that honour the data, particularly in areas of great uncertainty (Bond et al., 2007; Bond et al., 2015). Interpreters need to be able to identify the key elements (e.g. growth geometries, regional level) and employ different validation techniques (e.g. balancing or restoration) that allow*

*differentiating between (a priori similar) conceptual models (Bond, 2015). The conceptual models therefore incorporate all the elements that shape the knowledge of the geologist of a certain aspect of the geology; for example, the conceptual model of a turbidite system will include characteristics about their origin and evolution, common stratigraphic sequences, lithological composition, stratigraphic structures associated, etc. These conceptual models are dynamically modified or renewed with the arrival of new observations (input information), and are used to produce predictions (inferences) that can help to answer questions about the world (Shipley and Tikoff, 2017). Conceptual models are therefore the basis of the interpretation, as they provide the necessary criteria to make sense of the data (Frodeman, 1995).*

To deal with uncertainty, interpreters employ heuristics (or 'rules of thumb') in the process of generating the conceptual models, and that makes them subject to a broad range of cognitive biases (Kahneman et al., 1982). One of these biases is related to the capability of interpreters to adjust their interpretations from their initial ideas or conceptual models. This type of bias, called anchoring, has been identified in many decision-making processes since it was first described by Tversky and Kahneman (1974), and takes place in the seismic interpretation process. Rankey and Mitchell (2003) investigated the effect of anchoring in an interpretation experiment by asking interpreters to reassess their seismic interpretations after being provided with additional well data. Their work shows that most interpreters did not feel that their interpretations needed to change substantially, in spite of data showing changes in porosity and net-to-gross predictions that did not fit with their initial interpretations. Their results suggest that interpreters were anchored to their initial conceptual models, and that they were reluctant to change their mind in light of new data. In a different experiment, Bond et al. (2007) observed that participants asked for the geographical location of the section and suggested that interpreters could use this information to build their conceptual models, by using geographically specific knowledge of e.g. the relevant tectonic setting to anchor their interpretation. For example, an interpreter knowing a seismic section was from the North Sea may assume a conceptual model based on an extensional tectonic regime and will consciously and unconsciously look for normal faults in the seismic data. However, if the conceptual model is wrong, e.g. there is significant inversion in the seismic section, the interpretation could be compromised. *So although conceptual models can be dynamically modified or renewed with the arrival of new observations, as described by Shipley and Tikoff (2017) and others, anchoring bias often results in limited adjustment from initial models.* Thus, although conceptual models are needed to develop geologically sound interpretations, they can also create anchors to potentially erroneous *outcomes.*

In the discussion we now introduce a new figure (9) that shows the interpretation process. We agree with the reviewer that "horizons cutting reflections" may reflect a lack of understanding of seismic interpretation, but we do not believe this to be the case. All the students had experience in seismic interpretation and there were no models of left-ward dipping faults with a reverse sense of motion have been interpreted, in which horizons would very distinctively have cut seismic reflectors. We have added in a new sentence in the discussion on this and refer to a new figure (9d).

On experience more generally see response to comment 1. We have also added the following paragraph into the discussion:

*'Experience and knowledge are expected to have played a key role in informing the initial observations that led to selection of a conceptual model at the beginning of the interpretation. We purposely chose a student only cohort to mitigate against the competing effects of experience and knowledge with other factors we wanted to test. To ensure this was the case we have analysed the data for differences in interpretation outcome between students from different Universities and between undergraduate and postgraduate students. This analysis shows no strong evidence that experience had an effect on interpretation outcome.'*

3. Vertical exaggeration analysis. Although the exercise is related to the perception of the interpreter to the scale, the authors based their analysis on the interpretation of seismic sections in time. It seems that the authors consider 1:2 and 1:4 represent vertically exaggerated displays of a 1:1 section. These scales are more likely to be display factors for seismic sections in time. A 1:1 display factor in time will be significantly different from a 1:1 section displayed in depth. The same problem will apply for other display factors in time, which will not be representative of the real scale of the structures in depth. Moreover, the depth section will depend on the velocity model used for depthconversion. For example, assuming an unrealistic and simple model of a constant velocity of 5000 m/s throughout the section, the 1:2 display ratio you used will be equivalent to a 1:~0.7 depth ratio, whereas at a velocity of 4000m/s, this will be equal to

a 1:~0.9 depth ratio. This means the section with 1:2 display is more likely horizontally stretched and not vertically exaggerated. The analysis of the impact of vertical exaggeration should, therefore, be performed in a depth section and possible in PSDM section. I suggest you depth-convert a section, so you have a reference for what a 1:1 section in depth looks like. Then compare these to the time sections given to the students and analyse the result taking into account the depth section. You can also consider additional display factors (e.g. 1:6, 1:8) in time to complement your study.

The seismic section used was extracted from the Virtual Seismic Atlas ([www.vsa.org](http://www.vsa.org)) where it is stored with an approximately 1:1 display, as explained in the description of the section. Even if it was not the case, and the display was 1:1.3 (or 1:0.7) our results would be consistent, because we vertically exaggerated the original image 2 times and 4 times to use in the experiment, and then converted the interpretation back to the 1:1 (or 1:1.3-1:0.7) for analysis. We have added a sentence to make clear that the original image had no vertical exaggeration, according to the source of the data:

*The section used in this experiment was originally downloaded with no vertical exaggeration (i.e. with an approximate horizontal to vertical ratio of 1:1), according to the Virtual Seismic Atlas information.* In a series of interpretation experiments, this seismic image was presented to participants with horizontal to vertical exaggeration of 1:4 (Figure 2a) or 1:2 (Figure 2b), hereafter called *1:4* and *1:2* sections.

We agree that the velocity model has a strong impact in the interpretation outcomes. This issue was partly studied in an experiment by Alcalde et al. (2017), whose research shown that the depth conversion had a major impact on the interpretation, even if this was mostly related to the changes in the image quality derived from the depth conversion process. We have no data to inform velocity models for our seismic section to run a robust depth conversion, and therefore this option was dismissed during the experiment design phase.

Alcalde, J., Bond, C.E., Johnson, G., Ellis, J.F. and Butler, R.W.: Impact of seismic image quality on fault interpretation uncertainty. GSA Today, 2017.

Finally, on adding more display factors would be helpful to constrain the conceptual model vs vertical exaggeration issue. However, we did not include more displays due to the subject availability (160 students would have meant only ~40 participants per display, reducing the statistical meaning of the results), and to the display options (i.e. a 1:6 display would not fit an A4 sheet, or would require reducing the 1:2 section too much). Nevertheless, we have added the suggestion of Reviewer #2 to the result section, to acknowledge that using more displays would help to constrain the results:

Similarly for the right-ward dipping fault interpretations normal fault dip angles are low 24º-27º, but not as low as those interpreted to the right, suggesting that the angle of dip of the fault is driven more by the seismic image data than by any effects of vertical exaggeration. *Testing with more display options (e.g. 1:6 or 1:8 vertical exaggeration) could be helpful to confirm this finding, and would be interesting lines for further enquiry.*

4. Fault dip data. The author should clearly state in the text that the measurements were made for comparison and are not representative of real fault dips.

We have amended this issue in Section 3 to make this clear (interpretation results):

The interpretation results were digitised manually and then converted to a 1:1 vertical exaggeration (VE=1:1) for comparison; therefore, the fault dip angles presented in this work are VE=1:1 in time. *As the sections were interpreted in TWT, the analysed dips of the faults are not real dips (i.e. these observed in sections in depth), but their relative differences are still comparable.* Individual examples of the interpretation results after digitisation from both the *1:2* and *1:4* sections are shown in Figure 3.

We also added a table (new table 1 in the manuscript) with the median dip angles depth converted using a velocity of 3000 m/s (according to Stewart, 2011). This way we are able to compare the dip results with the Andersonian models in the discussion, and provide more realistic fault values than the ones calculated in TWT.

Table 1: median values in two-way traveltime and their depth-converted equivalent of the 1:2 and 1:4 sections, divided by dip direction and fault motion.

| Section | 1:2 | | | 1:4 | | |
|---|---|---|---|---|---|---|
| Dip direction | Right | Right | Left | Right | Right | Left |
| Fault motion | Normal | Reverse | Normal | Normal | Reverse | Normal |
| Median (TWT) | 13° | 23° | 16° | 10° | 21° | 22° |
| Median (depth-converted) | 19° | 33° | 23° | 14° | 31° | 30° |

The analyses of fault dips should be divided based on the conceptual models, as different assumptions were made for these. Mixing data from different conceptual models based on dip direction as in figures 4 & 6 seems inadequate. The authors mentioned that right-dipping reverse faults required higher angles than right-dipping normal faults so that both populations will differ due to the assumption made during the interpretation. Hence, these should be treated separately as in figure 5 & 7. The data shows significant variability suggesting that the average should not be used. In figure 8, the analysis of fault dips should take into account the position of the faults within the section. As they are, the results show too much dispersion and mix different conceptual models (right-dipping faults). It is not possible to correlate faults between different display scales. Hence, it is not possible to know if the perception changed due to the scale. As currently displayed (rose diagrams (Figure 7) and curves (Figure 8)), the results are difficult to interpret and should not be used. I suggest that the authors subdivide the dataset based on the conceptual models and use plots of horizontal distance vs fault dip. For example, you can compare the horizontal distribution of the interpreted faults to see which faults are recurrent in the interpretations, and are located in similar places. Then for each location where faults are repeated, you can analyse the "dip" distribution and calculate a representative value for that population. This can then be compared between scales and plotted in figure 8.

We generally agree with Reviewer #2 that the average values are not fully representative of the fault dips, given the skewness featured by some of the distributions. To solve this, we have substituted the average values with the median values, which, together with the standard deviation (already shown in figures 6 and 7) provides a clearer picture of the characteristics of the dip distributions. We have also modified Figure 8 to consider the medians instead of the average dip values, and have included the medians of the dip direction (left or right) and fault motion (normal or reverse) sub-cohorts for comparison. We have also amended the discussion in the text with the new median data. The median results do not produce significant changes in the overall results, but the suggestion of adding sub-cohorts to the analyses (particularly in Figure 8) has helped to disentangle the effects and interrelationship of vertical exaggeration and conceptual modelling in interpretation outcomes. With these additions to the text and the figures, we hope that the results are clearer to the reader.

We disagree with the suggestion to remove figures 4 and 6. Figures 4 and 6 show the preliminary split of the results by cohorts, to which the sub-cohorts presented in figures 5 and 7 belong (i.e. figures 5 and 7 are sub-cohorts of these in figure 4 and 6). We believe that the current structure of the methodology is the most appropriate to describe the rationale followed, both in terms of the steps followed and in the order they were applied: we first analysed the interpretations as a whole and identified the three potential cohorts (i.e. 1:2 vs 1:4, left dipping vs right dipping and reverse vs normal). After analysing these cohorts we split even more these cohorts into the cohorts presented in Figs 5 and 7, based on the hypothesis that fault dipping direction had more influence in the outcomes than vertical exaggeration, but this hypothesis can only be formulated once the overall results have been discussed. We see no reason to remove figures 4 and 6 from the manuscript.

Regarding the subdivision by faults, we chose not to further subdivide the results fault by fault, as this would only add even more complexity to the results and the population (i.e. the number of faults interpreted per faulted section) will be reduced dramatically, potentially below statistical significance. Reviewer #2 has already shown their concerns about the complexity of the results, but we do not believe that further splitting of the data into even more sub-cohorts (as many as the faults present in the seismic image, 5-10 times more than the current) will help in this matter. Furthermore, these results would be relevant if the two vertically exaggerated sections were presented to the same subject, and this was not the objective of the experiment. The seismic section was selected because it presented faulting in domino blocks, with little variability in fault dip. The presence of the variability in fault dip is captured within the SD and the rest of statistical analyses, and these are averaged out across the entire section. The differences in variability are already discussed in the text, in section 4.2 Fault dip variability, which has already modified as suggested by Reviewer #2.

- Final remarks Vertical exaggeration and anchoring are important aspects that should be taken into account during seismic interpretation. Research on the impact of these in the outcome of the interpretations can contribute to the seismic

interpretation workflow. However, the way the experiments are designed and the way results are present are crucial. Although the authors try to discuss the importance of vertical exaggeration and anchoring, the paper in its current state does not give support for their conclusions. The effect of anchoring is based on assumptions, and the existence of bias cannot be evaluated from the data and the analyses presented. The experiment, as described, is unlikely to support a discussion on anchoring and bias. The use of time sections makes the results uncertain, and their analyses do not include the background and experience of the interpreters. I suggest the methodology used to analyse the data needs to be modified. Current diagrams are difficult to analyse and sometimes combine the result of different conceptual models. Some of these do not support their discussion points. I suggest the author should revisit the methodology and parameters used to analyse the data, as well as the way the results are displayed. This inevitably requires major changes in the way the paper is presented, including significant modifications to the text and figures. The discussion and conclusions should be reconsidered after these modifications.

We are very thankful for the thorough review and comprehensive comments. We acknowledge that the methodology presented is not the perfect experiment, but in spite of the complexity of the study and the multiple inter-relationships, we have attempted to quantify the impacts. From this we believe that the data and analysis support discussion of the issues of interpreting vertically exaggerated seismic images and in conceptual model anchoring. We believe we have been open in qualifying the limitations to our study, and hope that it presents a starting point for future study in these areas. We have made significant changes to the manuscript according to the comments made by Reviewers #1 and #2 resulting in a clearer manuscript.

Specific comments in the supplementary file

Unless specifically mentioned below, we have introduced all changes suggested in the supplementary by the reviewer.

Section 1 – Introduction

Seismic images are indirect representations of  the physical properties of rocks in the subsurface.

Seismic images are indeed representations of the changes in physical properties of the rocks; if there were no changes, the seismic image would be blank. So we decided not to remove this from the text.

Consider putting together all the examples of the impact of vertical exaggeration on interpretation.

We do not understand this comment. The current structure presents positive examples of vertical exaggeration (paragraph 1) and after that examples where the vertical exaggeration disturb the perception of the interpreters and can hence lead to erroneous interpretations (paragraph 2). In any case, both groups of examples are one after another.

Are faults planar, listric, both? was this taken into account?

The participants' results included 70 interpretations (44% of the total interpretations) showing curved faults and 62 interpretations (39%) showing planar faults. However, the dip analysis was calculated in a single point (1.1 ms TWT) crossing at approximately the mid-point of all the faults, that we assume is a representative value for the whole of the fault, and this way the calculation is independent of the fault curvature.

It seems that this interpretation did not fulfil the requirement of marking a horizon. I suggest this is not included and discussed the reasons with the other interpretations that were not taken into account.

We use this single left-ward dipping reverse fault interpretation as a proof that this kind of interpretation is largely impossible. Interpreting the horizons was secondary with respect to interpreting the faults, which was really the main requirement.

I suggest you clearly divide the definition of the seismic unit and then how the conceptual model is created based on how the units were defined. For example, you can use subheadings like 1.1 seismic units, 1.2 conceptual model 1, 1.3 conceptual model 2 & 1.4 conceptual model 3). You should consider including your preferred model.

We do not agree with this comment. The units and conceptual models are already clearly separated and numbered. The discussion is also continuous, and we do not envisage how adding subsections will help the flow of the discussion. We also prefer not to include any "preferred model(s)", as we want to study the interpretation results objectively (i.e. we would rather avoid talking about "right" and "wrong" interpretations). Having a preferred model (and clearly stating this preference) would bias the reader towards this model, and could deviate the discussion from its original purpose.

Is this anchoring comparable to the one that you described in the introduction when additional data is given?

See response to question 2 above.

This paragraph should be in the discussion. However, this is not clear from your results.

We disagree with this comment. This paragraph summarises a recommendation (that awareness of biases is important) based on the fact that we found that interpretations were affected by anchoring bias. We do not think that this should be part of the discussion, as this is a further recommendation extracted from the results discussed previously.

---

## Author Response (AR2)

Following the Editor's suggestion, we have modified the title of the paper. The new title is:

Fault interpretation in seismic reflection data: an experiment analysing the impact of conceptual model anchoring and vertical exaggeration